# Point-of-Care Testing for SARS-CoV-2: A Prospective Study in a Primary Health Centre

**DOI:** 10.3390/diagnostics13111888

**Published:** 2023-05-28

**Authors:** Rob Daniels, Juliette Cottin, Nagham Khanafer

**Affiliations:** 1Townsend Health Medical Centre, Seaton EX12 2RY, UK; 2CEMKA, 92340 Bourg-la-Reine, France; juliette.cottin@cemka.fr; 3Department of Hygiene, Epidemiology, and Prevention, Lyon University Hospital and Centre International de Recherche en Infectiologie, 69007 Lyon, France; naghamkhanafer@hotmail.com

**Keywords:** COVID-19, implementation, point-of-care-test, primary care, SARS-CoV-2

## Abstract

Background: In 2020, health systems across the world responded to the COVID-19 pandemic by making rapid changes to reduce the risk of exposure in patients and healthcare professionals. The use of point-of-care tests (POCT) has been a central strategy in dealing with the COVID-19 pandemic. The aims of this study were to evaluate the impact of POCT strategy (1) on maintaining elective surgeries by removing the risk of delayed pre-appointment testing and turn-around times and (2) on time dedicated for end-to-end appointment and management, and (3) to assess the feasibility of using the ID NOW^®^ among healthcare professionals and patients in a primary care setting, requiring pre-surgical appointment and minor ENT surgery in the Townsend House Medical Centre (THMC), Devon, United Kingdom. Methods: A logistic regression was performed to identify factors associated with the risk of canceled or delayed surgeries and medical appointments. Second, a multivariate linear regression analysis was conducted to calculate changes in the time dedicated to administrative tasks. A questionnaire was developed to assess the acceptance of POCT in patients and staff. Results: 274 patients were included in this study; 174 (63.5%) in Group 1 (Usual Care) and 100 (36.5%) in Group 2 (Point of Care). Multivariate logistic regression showed that the percentage of postponed or canceled appointments was similar between the two groups (adjusted OR = 0.65, [95%CI: 0.22–1.88]; *p* = 0.42). Similar results were observed for the percentage of postponed or canceled scheduled surgeries (adjusted OR = 0.47, [95%CI: 0.15–1.47]; *p* = 0.19). The time dedicated to administrative tasks was significantly lowered by 24.7 min in G2 compared to G1 (*p* < 0.001). 79 patients in G2 (79.0%) completed the survey, and the majority agreed or strongly agreed that it improved care management (79.7%), decreased administrative time (65.8%), reduced the risk of canceled appointments (74.7%) and the traveling time to do COVID-19 test (91.1%). Having point-of-care testing in the clinic in the future seemed more than welcome by 96.6% of patients; 93.6% declared to be less stressed by having the test at the clinic than waiting for the results of the test realized elsewhere. The five healthcare professionals of the primary care center completed the survey, and all agreed that the POCT positively influences the workflow and can be successfully implemented into routine primary care. Conclusions: Our study shows that NAAT-based point-of-care SARS-CoV-2 testing significantly improved flow management in a primary care setting. POC testing was a feasible and well-accepted strategy by patients and providers.

## 1. Introduction

The coronavirus disease (COVID-19) pandemic began in Wuhan, China, in December 2019, and the outbreak rapidly spread worldwide. The pandemic increased the need for health services and led to radical changes in the delivery of health services [1].

Its impact extends far beyond the short-term burden on healthcare systems linked to COVID-19 infection, to include the adaptations that services and society have developed to reduce the risk of inadvertent spread [2]. The need to try to maintain usual health services as close to normal as possible while minimizing risk has demanded innovation and change.

Part of this change in the management of the COVID-19 pandemic has been the efficient development of rapid diagnostic tests that were complementary or, in some instances, as an alternative to the laboratory-based nucleic acid amplification test (NAAT) which is currently the reference standard diagnostic tool.

Point-of-care testing (POCT) has improved the ability of healthcare systems to rapidly identify and manage COVID-19 cases, as well as reaching remote areas where laboratory-based NAAT testing was not feasible [3]. The diagnostic performance of the Abbott ID NOW^TM^ for COVID-19 assay has previously been reported in many studies [4,5,6,7]. In 2020, the test was authorized as a lab-based and POC diagnostic assay for the detection of SARS-CoV-2. Therefore, the performance of ID NOW^TM^ was not assessed in this study. Compared to the traditional laboratory PCR, which provides results in average two days, the ID NOW COVID-19 2.0 is a rapid (positive results as early as 6 min, negative results in 12 min), instrument-based molecular isothermal nucleic acid amplification technology (NAAT) test for the qualitative detection and diagnosis of SARS-CoV-2 from nasal and nasopharyngeal swabs [8].

To the best of our knowledge, there are currently no rapid diagnostic tests that have been evaluated as fit-for-purpose in primary care to identify patients with COVID-19, as highlighted by a recent Cochrane review of 16 papers, including 7706 participants in total [9]. The high risk of selection bias, with many studies including patients who had already been admitted to the hospital or who presented to hospital settings with the intent to hospitalize, leads to findings that are less applicable to people presenting in primary care, who on average experience a shorter illness duration, less severe symptoms and have a lower probability of the target condition. Prospective studies in an unselected population presenting to primary care or hospital outpatient settings are needed [10].

During the covid-19 pandemic, patients being seen in an elective primary care Ear, nose and throat clinic would usually be informed of the appointment day and asked to do a PCR test two days before the presentation. It is common to have some delays in obtaining the results before the presentation day, or even because of logistics, and the test may not be performed. This poses the risk of cancellation, not the determined outcomes themselves. 

In addition, going to a laboratory to perform the test could expose the sample to a risk of contamination. In the meantime, the patient may also become infected through social mixing and contact with family members during the window of two days to obtain the result. Patients who test positive on a PCR or on a POCT test, have their appointment cancelled, and the time delay to arrange and receive the results of testing, prevents short notice usage of appointments. Patients who have POC testing with positive results will also have to cancel their clinic appointment. However, it would be easy to reschedule right there and then (with lesser administration time) since they already know their positive status, rather than still waiting to know the PCR test outcome where appointments/surgeries cannot be rescheduled until this is known.

In this context, it is important to assess the impact of the implementation of a POCT solution (Abbott ID NOW^TM^) for rapid diagnosis of COVID-19 in a primary care setting, offering otolaryngology/minor surgery services in the United Kingdom.

The THMC provides a pre-operative assessment, including COVID-19 status, for an average of 16 patients per week. On a weekly basis, an average of 16 patients are seen and reviewed for surgery. 

Patients on the G1 arm were notified about their appointments and required to do a PCR test two days before presentation. 

In the usual care prior to the study, the administrative time related to COVID-19 is described as the average operational time frame it took the staff to administer the task of notifying the patient (via email) to perform the test ahead of scheduled appointment day to coordinate the PCR test kits dispatch with the COVID-19 team, confirm the results and schedule the appointment/surgery depending on the test outcome. The mean time needed per patient is estimated to be 2 h. It took this length of time on average given that the staff is normally involved in other administrative work that is not related and dedicated to this service, e.g., secretary duties, meetings in between, lunch breaks, etc.

Currently, this status is a lab-based PCR test (Usual Care). This procedure is characterized by (1) a dedicated administrative time of approximately 30 h per week and (2) a risk of reduced activity due to canceled or postponed appointments. The implementation of the POCT solution allowing a diagnosis in less than 15 min, could improve this situation.

For patients included in G1, the procedure was the following:Consultations and tests are performed as usual (two days before), and the decision to maintain or postpone or cancel the surgery/appointment was made according to existing protocols upon receipt of laboratory results;As described above, for the usual care, evaluation of the administrative time related to COVID-19 (from the patient email notification to checking the results and rescheduling appointments as necessary) by a “chess clock” approach. The chess clock is the time stamp recorded for all administrative and operational activities to administer the task as described earlier. When the staff is engaged in some other activity non-related to the pilot, the chess clock goes off and is clicked back on upon resumption of administrative duty dedicated to the pilot.

However, for patients included in G2, the steps were:Consultations accomplished as usual;Test carried out directly by the THMC staff using ID NOW when the patient arrives on the scheduled appointment day. No emails or test kits are sent prior to the presentation. The decision to maintain or postpone or cancel the surgery/appointment was made according to existing protocols upon knowledge of the test result;Evaluation of the administrative time dedicated to COVID-19 (from checking the results to rescheduling appointments as necessary) by a “chess clock” approach (as described above);Surveys’ documentation.

The aims of this study were to evaluate the impact (1) on surgical appointments (the percentage of postponed/canceled appointments), (2) on time dedicated to administrative tasks, and (3) to assess the feasibility of using the ID NOW^®^ among THMC healthcare professionals and patients in integrating into patient pathways and care management through a questionnaire.

## 2. Materials and Methods

A prospective non-interventional study was designed to evaluate the impact of two modalities to test COVID-19 in 2 groups (G1 and G2) of patients requiring minor surgery in the Townsend House Medical Centre (THMC), Devon, United Kingdom. The study complied with the General Data Protection Regulation (GDPR) and was conducted in accordance with Good Clinical Practice.

The processing of the personal data of participants was minimized by making use of a unique participant study number only on all study documents and any electronic database(s). All documents were stored securely and only accessible by study staff and authorized personnel. The study staff safeguarded the privacy of participants’ personal data.

The study was conducted between February and June 2022, and patients were included in accordance with a sequence of 8 weeks for each group. The entire study duration was 16 weeks, with an 8-week alternate period between the 2 comparative groups. Since the mean number of weekly consultations was 16 and a prevalence “drop off” of 20%, the minimum number of patients necessary for the investigation was estimated to be 100 in each group. Each patient recruitment and allocation to comparative groups were contingent on the presentation on an alternate week basis as prescribed in the study design (Usual Care on/off).

As such, the week the patients presented would determine the arm of the study they were allocated to. The study kicked off with the patients presenting in Week 1 allocated to the G1 (Usual Care Arm) and subsequently on an alternate 2-weekly basis for other patient appointments for the study duration. Patients presenting at Week 2 were allocated to the G2 (POCT) arm and placed on an alternative 2 weekly appointments. The term “Usual Care” means the current process at THMC, based on the necessity to obtain PCR results upon the appointment, and the term “POCT” refers to the process evaluated in this study, based on results obtained directly at the consultation sites.

Summarily, patients in the POCT arm were tested on the day of the scheduled appointment, and in the PCR arm, prior notification of the scheduled appointment with PCR test kits was sent to patients 2 days prior to appointment day. The study design and procedures are also summarized in Figure 1. In patients with negative PCR results, the appointment is the expected plausible outcome. However, in some conditions (e.g., problem resolved, patient’s decision), the appointment might be postponed or cancelled irrespective of the negative results.

### 2.1. Study Population

Patients were recruited in the pre-operative consultation at the THMC. The inclusion criteria were (1) adult patients over 16 years, (2) attending THMC for ENT outpatient clinic, and (3) having nasal/nasopharyngeal swab for COVID-19 as part of clinical care at THMC. Nasal and NP swabs are the only on-label allowable swabs to be taken as recommended by the manufacturer’s instructions and performed by the patient irrespective of the comparative arm they belong to, and (4) able to give informed consent for participation in the study. However, (1) adults unable to understand the study information and to give consent, and (2) who needed immediate hospitalization were not allowed to participate in this study. 

Informed consent is required to participate in the study only and not for the consultation. THMC has dedicated processes in place, which align with the ethical principles, conduct, and professionalism that support good clinical practice for participants that fall into this category.

The healthcare professional in the department of pre-operative consultation of the THMC was invited to participate in the acceptance survey. Participation was limited to those who had given consent.

### 2.2. Data Collection

The following data were completed for each participating patient: date of inclusion (=date of the appointment), age, gender, past medical history (cardiovascular, respiratory or psychiatric diseases, chronic renal failure needing dialysis, diabetes, immunosuppression, and obesity with a body mass index over 30), COVID-19 history (symptoms on the day of appointment, recent contact with COVID-19 cases, specific treatment for COVID-19 and vaccine status) and the results of PCR or POCT for COVID-19. 

Data related to the organization were the availability of the test and associated time (in minutes). This includes PCR performed or not, time to check the results, and informing of COVID-19 team of results for PCR or POCT tests. The administrative time per patient was estimated according to the following variables: Time to send an e-mail or phone to notify the patient within 2 days before the appointment, the time to send an e-mail or phone the medical laboratory to obtain the results, the time to discuss with patients if the surgery needs to be postponed and the time to reschedule a new appointment. The impact of each strategy on appointments was evaluated with the following variables: the number of canceled or postponed assessments and the reasons for the cancellation of surgery.

### 2.3. Statistical Analysis

Descriptive analysis of continuous variables was summarized as mean, minimum and maximum, interquartile, and as frequency and percentage calculated on the expressed responses for qualitative variables. The normality of the quantitative variable was assessed using a Shapiro–Wilk test, and they were compared using Student or Wilcoxon or Kruskal–Wallis tests, while categorical variables were compared using Fisher’s exact test. 

A logistic regression was performed to identify the factors associated with the risk of canceled or delayed appointments across G1 and G2 groups. Second, a multivariate linear regression analysis was performed to calculate changes in the time dedicated to administrative tasks. 

Results were expressed as odds ratio (OR) and 95% confidence intervals (CI 95%). All *p*-values were two-tailed. *p * < 0.05 were considered statistically significant. Statistical analyzes were conducted with R software version 4.2.1.

## 3. Results

A total of 274 patients were included in the analysis: 174 (63.5%) in G1 and 100 (36.5%) in G2. The mean age of patients was significantly higher in G2 than in G1 (66.9 years vs. 59.5 years, *p* = 0.003). In G2, there were 55.0% females compared to 52.9% in G1. The frequency of comorbidities was significantly higher in G2 than in G1 (18.0% vs. 2.3%, *p* < 0.001), and the majority of patients had received three doses of the COVID-19 vaccine. The characteristics of included patients are described in Table A1. 

At inclusion, only two patients had been experiencing COVID-19 symptoms for at least 8.5 days; both were in G1. Five patients (5.0%) in G2 and nine patients (5.2%) in G1 reported contact with a confirmed case of COVID-19.

The appointment was postponed or cancelled for five patients (5.0%) in G2 and 17 (9.8%) in G1 with no significant difference between the two groups (crude OR = 0.49, [95%CI: 0.17–1.36]; *p* = 0.24). 

The percentage of postponed or canceled appointments remained not different between the two groups after adjustment on the following potential confounding factors: age, gender, comorbidities, vaccination status, prior COVID-19 treatment or symptoms, and exposure to confirmed COVID-19 case (adjusted OR = 0.65, [95%CI: 0.22–1.88]; *p* = 0.42). 

The scheduled outpatient consultations were postponed or cancelled for four patients (4.0%) in G2 and eighteen (10.3%) in G1 with no statistically significant difference between the two groups (crude OR = 0.36, [95%CI: 0.11–1.10]; *p* = 0.10). These percentages remained non-significant after adjustment on age, gender, comorbidities, vaccination status, prior COVID-19 treatment, COVID-19 symptoms, and exposure to confirmed COVID-19 cases (adjusted OR = 0.47, [95%CI: 0.15–1.47]; *p* = 0.19). 

The reasons for the eighteen canceled outpatient consultations in G1 were, for the majority of them, not related to COVID-19. However, COVID-19 was the cause of postponed or cancelled outpatient consultations for four patients in G2. These data are described in Table A2.

The time dedicated to administrative tasks was significantly lower in G2 compared to G1 (0.2 min vs. 25.1 min, *p* < 0.001), as shown in Figure 2 and Figure 3.

Recall that using the ”chess clock” methodology, the time dedicated to the administrative task is the time stamp recorded for all administrative and operational activities related to the pilot to administer the task beginning notifying the patient to perform the test to the point the results are received. The decision is made to cancel or proceed with the appointment/surgery for G1. For G2, it will only be the administrative time required to check the result and schedule an appointment since the patient walks in on the day of the appointment to take the test.

Multivariate analysis showed that after adjustment on different factors, the administrative time was 24.7 min lesser for patients in G2 compared to those in G1 (*p* < 0.001). For patients who presented symptoms related to COVID-19, an excess time of 6.6 min seems needed to manage their file compared to patients free of symptoms. However, the difference was not statistically significant (*p* = 0.20). The time was lower for patients vaccinated with at least two doses, but it was not statistically significant. The complete results are shown in Table A3.

Seventy-nine patients in G2 (79.0%) completed the survey (results shown in Figure A1), and the majority agreed or strongly agreed that it improved care management (79.7%), decreased administrative time (65.8%), reduced the risk of cancelled appointments (74.7%) and the traveling time to do COVID-19 test (91.1%). Most of the participating patients disagreed or strongly disagreed with receiving a postal test kit and dropping it back at their chosen hospital (82.3%). The majority of participants (92.4%) also disagreed or strongly disagreed with travelling to take a PCR test elsewhere. Having point-of-care testing in the clinic in the future seemed more than welcome by 96.6% of patients, and 93.6% declared to be less stressed by having the test at the clinic than waiting for the test results realized elsewhere. The complete results of the survey are presented in Figure A1.

The five healthcare professionals of the THMC completed the survey, with results shown in Figure A2. All of them (100%) agreed or strongly agreed that POCT is easy to use and to integrate into the patient pathway, improves care management and convenience with hygiene rules, reduces the number of cancelled/postponed procedures, and reduced the administrative time and patient burden of traveling elsewhere for testing. They also reported they would continue using POCT in the future. 

One of the HCW participants (20%) declared to be undecided about the role of POCT in facilitating medical decisions about procedures, and reducing the fear of being contaminated, the risky practices when isolation is needed and the time of sick leave due to contagion.

## 4. Discussion

The exponential growth of new COVID-19 cases overwhelmed healthcare systems worldwide and resulted in a disruption of the healthcare provision across different medical specialties [11]. While hospital care was receiving much attention, primary healthcare services suffered from a reduction in their allocated resources [12] and were undermined by the emerging measures despite their role as the first contact of patients with the healthcare system. New guidelines have been adopted worldwide to ensure the safe admission of patients into hospitals and to keep patients, staff, and visitors at bay from exposure to the infection. They included COVID-19 screening before hospital admission, which led to an increase in postponed or canceled surgeries due to positive tests. These rules affected medical and surgical specialties as described in some investigations, mainly in academic settings [13].

In addition, this management was severely hampered by the long delays associated with the centralized laboratory PCR testing, which often took several days to generate results. Point-of-care testing is associated with a significant reduction in the time needed to obtain the PCR results and could lead to improvements in infection control measures and patient flow compared with centralized laboratory PCR testing [4,7,14].

Since the beginning of the pandemic, THMC had spent more time in pre-clinic assessment due to the rules concerning COVID-19 status, due to the high risk nature of the nasendoscopy and instrumentation of the pharynx and middle ear in some patients. Currently, the procedure is a lab-based PCR test and is characterized, in addition to the risk of reduced activity due to canceled or postponed appointments, by the need to dedicate one equivalent full-time employee to manage administrative tasks in relation to COVID-19 tests. In response to these difficulties, we established a project to assess the role of a POCT solution (Abbott ID NOW^TM^) in accelerating the process by allowing a diagnosis in less than 15 min. In addition, we asked the patients and practice staff about their acceptance of the POCT solution.

Overall COVID-19 positivity in the included patients was low throughout the study period, which included one year after the large campaign of vaccination in the UK and during a period of a relatively low incidence rate, especially between April and June 2022 [15]. Additionally, almost all of the included patients had received at least two doses of the vaccine, which is known to be associated with a reduced number of new COVID-19 cases [16]. However, asymptomatic carriers can still transmit SARS-CoV-2, which is a major impetus for continuing universal pre-procedural testing [17].

The only variable that had a significant impact on a patient’s administrative time was whether they were in G2. In fact, our results showed that a patient in this group required almost 25 min less than a patient in G1. Routine molecular POCT for SARS-CoV-2 was associated with a reduction in time to results and time spent in admission cohort areas compared to laboratory PCR [18]. An active transition from routine diagnostic laboratories to the realm of high-sensitivity molecular diagnostics can significantly increase the efficiency and responsiveness of POCT and facilitate the management of COVID-19 in multiple settings [19].

To limit the risk of contracting COVID post-test collection, patients and members of their household need to reduce infection risk during the time window between laboratory tests and the appointment. Patients were advised to stay home after returning the sample kits to the laboratory and await the scheduled appointment days. A POCT performed on the day of the appointment could, therefore, reduce the risks associated with patients not following these instructions. As there were no means to monitor this, the best surrogate adopted was the correlate of positive testing outcomes in both groups. There were five positive cases (27.8%) in G1 and none in G2.

As an exploratory observation, this further entrenches the value of true onsite POC testing with Abbott ID NOW^TM^ for COVID-19 assay in limiting infection spread as the patients do not require waiting for two or more days for the appointment, with the attendant risk of contracting and/or contaminating other family members due to contact. It also eliminates the risk of infection spread to Townsend staff by the time they present on their scheduled appointment days and undergo aerosol generating procedures without the knowledge that they are positive on the day, even though their PCR results were negative from two days earlier. This was strongly demonstrated with the testing methodology for the POCT group, which tests on the appointment day and shows a more accurate picture of the infective status of the patient in real-time testing.

In this prospective primary-care study, we also assessed the acceptability of POC testing for COVID-19 in patients and staff members. To strengthen the evidence on social, psychological, or organizational consequences of POCT, it is essential to investigate the perspectives of the test users (staff) and patients on whom the tests are performed.

The attitude of patients is a significant factor in the successful implementation of such diagnostic procedures in routine healthcare [20]. Our study shows that patients accepted and appreciated the characteristics of the POCT, which provided them with an efficient alternative as they no longer needed to travel to do their test in a laboratory. Moreover, the immediate availability of the test results led to adjustments in patients’ behavior and emotional well-being. A similar result was reported in a German study conducted in primary care practices [21]. The authors of this cross-sectional survey conducted in 2021 in 13 primary care practices reported that patients highly appreciated the immediate availability of the rapid test result [21].

The healthcare professionals of the THMC declared that POCT is easy to use, integrated into the patient pathway and can improve care management. It is true that the number of participants is small, but it is still representative since the five professionals working in this center have participated in this survey. In a multicentric study, most of the participating general practitioners and medical assistants perceived POCT as easy to integrate into the daily routine of primary care and would use it, especially in situations with an immediate need for action [21].

Our study is the first to evaluate the impact of using POCT to reduce administrative time and the acceptance of patients and staff for molecular point-of-care SARS-CoV-2 testing in an English primary care center. One strength of our study is that the test was evaluated in a clinical routine. Patients completed the questionnaire immediately after rapid SARS-CoV-2 testing, which reduces the risk of recall bias.

Limitations in our study include being monocentric and only reporting the opinions of staff and patients in Devon, United Kingdom, although it is assumed that opinions from patients would be a reflection of the national attitudes at that time. Other limitations include a relatively small sample size, a short study assessment window, and only being performed in one area of the UK. A further limitation is this study was performed in a particular setting at a certain time.

Nevertheless, the unique situation of the pandemic has accelerated the movement of POCT, especially molecular POC testing in outpatient settings. Future studies to support the findings in this survey should assess both medical and economic factors for POCT in future respiratory seasons to fully assess the benefits of rapid molecular diagnostics and their place in general practice. These would deliver a more widespread acceptance of molecular POC at the GP level, which is required to improve the overall healthcare system regimes during these difficult annual seasons.

Studies in other countries or settings in the UK are necessary to confirm the generalization of our results. Differences in the healthcare system as well as in culture-specific attitudes toward healthcare, should be considered. Additionally, it is possible that patient perspectives on COVID-19 tests are affected by the pandemic situation and testing guidelines at the time. Almost all participating patients has been vaccinated, and the effect of vaccination status on the participants’ perspectives cannot be determined. Of note, no cost analysis was conducted for this study. No data on cost, such as labor and facility costs to run the testing, nor indirect costs to the patient were available. Data on other respiratory tract infections, such as influenza virus infection, suggest major cost savings by POCT, and it will be interesting to obtain more information evaluating cost analyses of such strategy for COVID-19 [22].

The pandemic has evolved, and this study period excluded the most recent Omicron variant outbreak and data on patients who have received their booster vaccines. As variants of COVID continue to emerge, policies for COVID testing will need to remain accessible and flexible to maximize the benefit and efficacy of the procedure.

## 5. Conclusions

The spread of COVID-19, leading to a global health crisis, has led to major disruptions in performing surgeries and other procedures requiring anesthesia. Despite the availability of vaccinations, the epidemiological burden of COVID-19 is still considerable, hence the routine preoperative COVID testing [17].

As the pandemic continues to evolve, protocols for COVID-19 testing for surgical procedures should be periodically re-evaluated. The results of the present study showed that the POCT strategy significantly reduced the administrative time in primary care structures offering a pre-operative assessment while being well accepted by patients and practice staff.

The pandemic remains active with successive waves, and the use of POCTs for SARS-CoV-2 detection will continue to be an important reference in both healthcare settings and in the community. However, the quality of POCT results needs to be monitored regularly, particularly in view of the emergence of new SARS-CoV-2 variants [3].

## Figures and Tables

**Figure 1 diagnostics-13-01888-f001:**
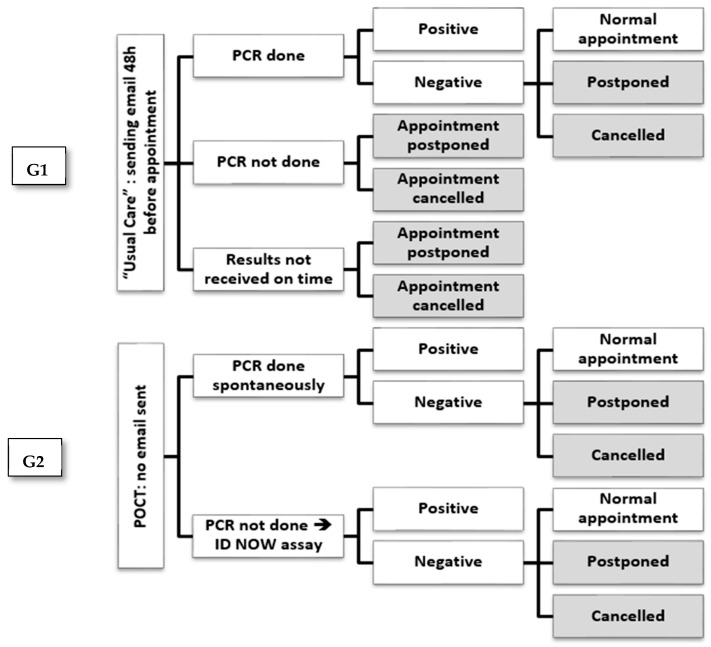
Study design and procedures for both study arms.

**Figure 2 diagnostics-13-01888-f002:**
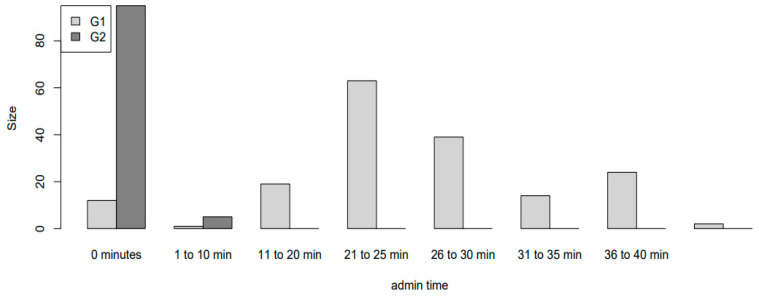
Mean administration time for both study arms.

**Figure 3 diagnostics-13-01888-f003:**
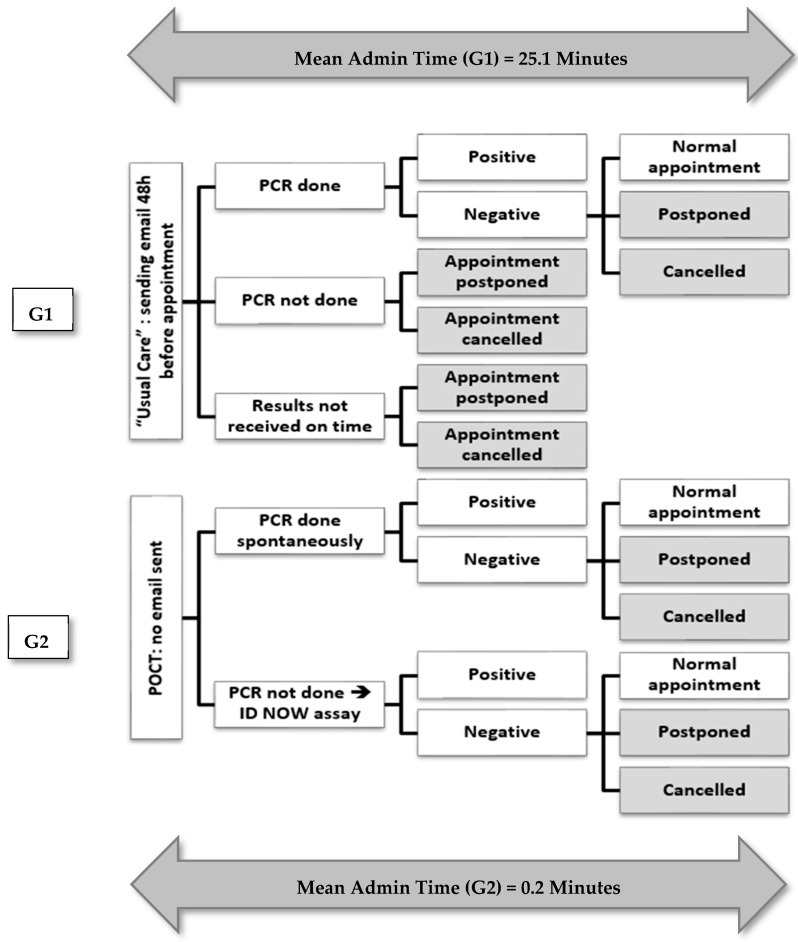
Study design and procedures with mean admin time results for both study arms.

## Data Availability

The data are not publicly available due to data privacy restrictions in relation to information on the health of the subjects.

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
