# Peer review of "Point-of-Care Testing for SARS-CoV-2: A Prospective Study in a Primary Health Centre"

_diagnostics, 2023, doi:10.3390/diagnostics13111888_

Round 1

Reviewer 1 Report

This paper describes the advantage of POCT over traditional NAAT for SARS-CoV-2 detection in patients. The major points of this paper is to address that POCT saves time and enhance efficiency for patients' appointments.

Some minor revisions on grammar and expression may be done to further improve the quality.

May provide a scattered dot/bar chart in your main figure for each patient's time taken in G1 and G2.

Discussion part is too lengthy and needs to be more concise.

Other points are attached in the attachment.

See attachment

Author Response

Point 1: Some minor revisions on grammar and expression may be done to further improve the quality.

Response 1: Corrected. Please find attached the revised manuscript.

Point 2: May provide a scattered dot/bar chart in your main figure for each patient's time taken in G1 and G2

Response 2: Completed. Kindly refer to Figure 2 (Page 7)

Point 3: The discussion part is too lengthy and needs to be more concise.

Response 3: Adjusted. Please find attached the revised manuscript.

Point 4: Other points are attached in the attachment.

Response 4: Changes completed. Please find attached the revised manuscript.

Reviewer 2 Report

I have some comments:

1- The author could further elaborate on the study's specific research questions and objectives. While the three aims are briefly mentioned at the end of the introduction, providing more detail on what the study aims to achieve and how it will contribute to the existing literature on COVID-19 testing would be helpful.

2- The language and writing style is generally clear and concise. However, there are some instances where the sentences could be rephrased or simplified for better clarity and readability. For example, in line 45, "acute infection with COVID-19" could be changed to "having COVID-19."

3- It is not clear from the methods section how patients were allocated to each group. Was it random or based on some other criteria? Please clarify.

4- The author mentions that the study complied with GDPR, but it is not clear how this was ensured. Were any specific measures taken to protect patient data beyond assigning a unique identifier? Please provide more details.

5- It would be helpful to know the characteristics of the patients in each group (e.g. age, gender, comorbidities) and whether they were similar between groups. This information would aid in the interpretation of the study results.

6- The author states that logistic regression was used to identify factors associated with the risk of canceled or delayed appointments, but it is not clear what variables were included in the model. Please provide more information.

 Minor editing of English language required

Author Response

Point 1: The author could further elaborate on the study's specific research questions and objectives. While the three aims are briefly mentioned at the end of the introduction, providing more detail on what the study aims to achieve and how it will contribute to the existing literature on COVID-19 testing would be helpful

Response 1: The aims of this study, as highlighted in the abstract, were to evaluate the impact of POCT strategy (1) on maintaining elective surgeries by removing the risk of delayed pre-appointment testing and turn-around times and (2) on time dedicated for end-to-end appointment and management, and (3) to assess the feasibility of using the ID NOW® among healthcare professionals and patients in a primary care setting, requiring pre-surgical appointment and minor ENT surgery in the Townsend House Medical Centre (THMC), Devon, United Kingdom

Relative to other existing literature, there were no clinical studies earlier published on the subject in evaluating the usability and the strategic clinical utility of POCT in removing clinical uncertainties to support patient management. Please refer to the statement in line 65, page 2.

‘To the best of our knowledge, there are currently no rapid diagnostic tests that have been evaluated as fit-for-purpose in primary care to identify patients with COVID-19 as highlighted by a recent Cochrane review of 16 papers including 7706 participants in total9

9Dinnes J, Deeks JJ, Berhane S, et al. Rapid, point-of-care antigen and molecular-based tests for diagnosis of SARS-CoV-2 infection. Cochrane Database Syst Rev. 2021;3[3]:CD013705. Published 2021 Mar 24

Point 2: The language and writing style is generally clear and concise. However, there are some instances where the sentences could be rephrased or simplified for better clarity and readability. For example, in line 45, "acute infection with COVID-19" could be changed to "having COVID-19."

Response 2: Corrected. Please find attached the revised manuscript.

Point 3: It is not clear from the methods section how patients were allocated to each group. Was it random or based on some other criteria? Please clarify

Response 3: Kindly refer to line 154 (Page 4).

‘Each patient recruitment and allocation to comparative groups were contingent on the presentation on an alternate week basis as prescribed in the study design (Usual care on/off).

As such, the week the patients presented would determine the arm of the study they were allocated to. The study kicked off with the patients presenting in Week 1 allocated to the G1 (Usual Care Arm) and subsequently on an alternate 2-weekly basis for other patient appointments for the study duration. Patients presenting at Week 2 were allocated to the G2 (POCT) arm and also placed on an alternative 2 weekly appointments’ – Page 4, Line 150 - 158

Point 4: The author mentions that the study complied with GDPR, but it is not clear how this was ensured. Were any specific measures taken to protect patient data beyond assigning a unique identifier? Please provide more details.

Response 4: Beyond the unique identifier, the patient data collected was restricted to age, sex, and co-morbidities (if present), as was necessary to demonstrate the demographic differences between the 2 comparative arms. The additional information on the indication for surgery was also provided and was relevant in contributing to the statistical differences of the test outcomes between POCT and the Usual Care (PCR), as some indications required follow-up and others did not require an appointment or surgery due to resolution.

As the data is being used for research purposes only, it will only be available in this paper ONLY unless subsequent studies can be used from them. To ensure incorrect data was NOT updated, data entries were performed by two independent persons into a data entry system and entries were verified and checked by a third person.

During the initial discussion about participation in the study, patients were provided with a comprehensive synopsis of the study and how data would be used, and that participation could be pulled at any time. No patient requested to be withdrawn from the study

Furthermore, as stated in the manuscript, all documents were stored securely and only accessible by study staff and authorized personnel. Confidentiality and safeguarding of the privacy of participants’ personal data was also strictly maintained.

Point 5: It would be helpful to know the characteristics of the patients in each group (e.g. age, gender, comorbidities) and whether they were similar between groups. This information would aid in the interpretation of the study results

Response 5: Kindly refer to Tables 1 and 3. (Appendix A, Pages 13-14)

Point 6: The author states that logistic regression was used to identify factors associated with the risk of canceled or delayed appointments, but it is not clear what variables were included in the model. Please provide more information

Response 6: All variables were used in the calculation of the propensity score, which was used to compare the 2 groups while adjusting for covariates that could have biased this comparison. The propensity score was calculated by using a logistic regression, based on the following factors associated with the risk of cancelled or delayed appointment: age, gender, comorbidities, vaccination status, prior covid treatment, symptoms, and exposure.

Reviewer 3 Report

The author in this manuscript has presented the implementation and acceptability of NAAT based Point-of-care (POCT) testing from Abbott (ID NOW) in a primary health care set up to effectively reduce the time dedicated to administrative task. The POCT reduced administrative time of health care professionals and waiting time of patients to get COVID-19 PCR result prior to appointment. The authors have also raised concerns over the quality and availability of Abbott ID NOW test for newly emerging SARS-CoV-2 variants. The observational study is relevant to hospital set up and can be implemented. The manuscript is well written. Few minor comments are as below:  

1. Introduction about Abott diagnostic POCT kit needs to be elaborated.

2. Line no. 22: In the abstract, while showing results the author need to specify G1 and G2 groups.

3.Few typographical/formatting errors need fixing. e.g. line nos. 83 and 84, grammatical error; line 110, double full stop; Line no. 212, extra space; Line nos. 287 and 331, Covid-19 needs to be in caps.

The writing is okay. However need minor corrections.  

Author Response

Point 1: Introduction about Abott diagnostic POCT kit needs to be elaborated.

Response 1:  Please refer to line 59, page 2.

‘Compared to the traditional laboratory PCR which provides results in average 2 days, the ID NOW COVID-19 2.0 is a rapid (positive results as early as 6 minutes, negative results in 12 minutes), instrument-based molecular isothermal nucleic acid amplification technology (NAAT) test for the qualitative detection and diagnosis of SARS-CoV-2 from nasal and nasopharyngeal swabs8‘

8globalpointofcare.eifu.abbott

Point 2: Line no. 22: In the abstract, while showing results the author need to specify G1 and G2 groups.

Response 2: Completed. Please find attached the revised manuscript.

Point 3: Few typographical/formatting errors need fixing. e.g. line nos. 83 and 84, grammatical error; line 110, double full stop; Line no. 212, extra space; Line nos. 287 and 331, Covid-19 needs to be in caps

Response 3: Corrected. Please find attached the revised manuscript.

Reviewer 4 Report

Rob Daniels submitted to Diagnostics an article on a study performed at local level in a primary health centre, focusing to the point-of-care testing for SARS-CoV-2.

The study appears to be described in detail and the sole author has declared the absence of conflict of interest with the Company producing and providing the diagnostic test.

Here are my suggestions for improvement:

- please, cite the references according to the Journal rules (e.g. in square brackets);

- LL 135-137: the concept of GDPR is mentioned twice;

- has the Declaration of Helsinki been taken into account? These aspects need to be made explicit;

- it is necessary to better explain the limits of this study (small sample size, limited territorial area, etc.) with respect to the features already briefly decribed;

- it is essential to improve the bibliography, implementing discussions with other studies that compare in detail your proposal, with other diagnostic methods, specifically highlighting their pros and cons;

- section relating to Authors Contribution: must be cited according to the Journal Rules, with the name of the Author and not of the Company producing the kit.

The English style used is fine.

Author Response

Point 1: Please, cite the references according to the Journal rules (e.g. in square brackets);

Response 1: Corrected. Please find attached the revised manuscript.

Point 2: LL 135-137: the concept of GDPR is mentioned twice

Response 2: Corrected. Please find attached the revised manuscript.

Point 3: Has the Declaration of Helsinki been taken into account? These aspects need to be made explicit

Response 3: Yes. All the principles (both basic and operational) and ethical standards in line with the Helsinki Declaration were considered and measures as described in the manuscript were taken to protect the interests, health, safety, and rights of the patient. A careful assessment of the risks was also done, which was non-existent in this evaluation. Informed consent was obtained and vulnerable groups were excluded as explained in the exclusion criteria. Operational protocols regarding the instrumentation and workflow as prescribed by the manufacturers in the users’ manual were strictly adhered to.

Local regulations and industry codes were taken into consideration including the UK (MHRA) GCP Guidance and Inspections for Clinical Trials, as well as the UK ABPI Codes.

Good clinical practice for clinical trials - GOV.UK (www.gov.uk)

Point 4: it is necessary to better explain the limits of this study (small sample size, limited territorial area, etc.) with respect to the features already briefly described.

Response 4: Limitations in our study include being monocentric and only reporting the opinions of staff and patients in that area; although it is assumed that opinions from patients would be a reflection of the national attitudes at that time. Other limitations include a relatively small sample size, a short study assessment window and only being performed in one area of the UK. A further limitation is this study was performed in a particular setting at a certain time.

Nevertheless, the unique situation of the pandemic has accelerated the movement of POCT and especially molecular POC testing in outpatient settings. Future studies to support the findings in this survey should assess both medical and economic factors for POCT in future respiratory seasons to fully assess the benefits of rapid molecular diagnostics and their place in general practice. These would deliver a more widespread acceptance of molecular POC at the GP level which is required to improve the overall healthcare system regimes during these difficult annual seasons

(Response inserted into the revised manuscript)

Point 5: It is essential to improve the bibliography, implementing discussions with other studies that compare in detail your proposal, with other diagnostic methods, specifically highlighting their pros and cons

Response 5:

Relative to other existing literature, there were no clinical studies earlier published on the subject in evaluating the usability and the strategic clinical utility of POCT in removing clinical uncertainties to support patient management. Please refer to the statement in line 65, page 2.

‘To the best of our knowledge, there are currently no rapid diagnostic tests that have been evaluated as fit-for-purpose in primary care to identify patients with COVID-19 as highlighted by a recent Cochrane review of 16 papers including 7706 participants in total1

1Dinnes J, Deeks JJ, Berhane S, et al. Rapid, point-of-care antigen and molecular-based tests for diagnosis of SARS-CoV-2 infection. Cochrane Database Syst Rev. 2021;3[3]:CD013705. Published 2021 Mar 24

Point 6: Section relating to Authors Contribution: must be cited according to the Journal Rules, with the name of the Author and not of the Company producing the kit

Response 6: Corrected. Please find attached the revised manuscript.